# Knowledge, attitude, and practice of Bangladeshi urban slum dwellers towards COVID-19 transmission-prevention: A cross-sectional study

**Md. Zahid Hasan**[1,2]*, **A. M. Rumayan Hasan**[1], **Md. Golam Rabbani**[1], **Mohammad Abdus Selim**[1], **Shehrin Shaila Mahmood**[1]

**1** Health Economics and Financing Research Group, Health System and Population Studies Division, icddr,b, Dhaka, Bangladesh, **2** Leeds Institute of Health Sciences, University of Leeds, Leeds, United Kingdom

* md.zahid@icddrb.org

**Data Availability Statement:** All relevant data contributing to the findings are within the study. The deidentified dataset of this study are available

## Abstract

The first COVID-19 case in Bangladesh was detected on March 8, 2020. Since then, efforts are being made across the country to raise awareness among the population for preventing the spread of this virus. We aimed to examine the urban slum dwellers' knowledge, attitude, and practice (KAP) towards COVID-19 transmission-prevention. A phone-based cross-sectional survey was conducted in five slums of Dhaka City. Total 476 adult slum dwellers were interviewed between October 31 to December 1, 2020 using a pre-tested questionnaire. During an interview, information was collected on participants' demographic characteristics and KAP items towards COVID-19. We used quartiles for categorization of knowledge and practice score where the first quartile represents poor, the second and third quartiles represent average while the fourth quartile represents good. Attitude score was standardized using z-score and identified as positive and negative attitude. Multiple linear regression models were used separately to identify the socioeconomic predictors of the KAP scores. The results showed that 25% of the respondents had good knowledge and 25% had poor knowledge, 48% had a positive attitude and 52% had a negative attitude, and 21% maintained good practice and 33% maintained poor practice towards COVID-19 transmission-prevention. About 75% respondents relied on television for COVID-19 related information. Regression results showed that knowledge and attitude scores were significantly higher if respondents had primary or secondary and above level of education compared to the uneducated group. Female respondents maintained significantly good practice compared to their male counterparts (β = 6.841; p<0.01). This study has found that one third of the studied slum dwellers maintained poor practice and one fourth had poor knowledge towards COVID-19 transmission-prevention. As KAP domains are significantly correlated, efforts are needed to raise awareness of COVID-19 particularly targeting individuals with average and lower knowledge to improve attitude and practice for the prevention of COVID-19.

as the Supporting information of the manuscript (S1 Data).

**Funding:** This study was funded by the Swedish International Development Cooperation Agency (Grant No# GR-01455). SSM was the main recipient of the grant. The funders had no role in study design, data collection and analysis, decision to publish, or preparation of the manuscript.

**Competing interests:** The authors have declared that no competing interests exist.

## Introduction

The coronavirus disease 2019 (COVID-19), a highly infectious disease, was first detected in Wuhan, China, at the end of 2019 [1]. The disease has spread rapidly across many countries and due to its terrible consequences, in March 2020, the World Health Organization (WHO) declared COVID-19 as a global pandemic [2]. In Bangladesh, the first COVID-19 case was detected on March 8, 2020, and the first person died of COVID-19 on March 18, 2020 [3]. By December 2021, Bangladesh reported more than 1,580,872 positive cases with 28,047 deaths from COVID-19 [4].

Being a highly infectious disease, COVID-19 can be transmitted in multiple ways such as being in close proximity to an infected person or being in an environment where droplets are generated from coughs, sneezes, or exhalation of an infected person; or by touching a contaminated surface, among others [5]. Though vaccines are now available, the risk of infection will remain until a large proportion of the population is vaccinated across the globe [6]. Therefore, controlling the spread of the virus with a behaviour change intervention related to transmission prevention is still considered as the most effective measure to protect people from the disease in the absence of pharmaceutical interventions [7]. Such interventions can guide how best to promote adherence to individuals' personal protective behaviours (individual behaviours aimed at protecting oneself and others). However, different enacted personal protective behaviours require different types of interventions guided by various behaviour change principles [8]. The capability, opportunity, motivation, and behaviour (COM-B) model can be implemented to change the behaviours needed to limit COVID-19 transmission e.g., cough or sneeze etiquette, using face masks, keeping physical distance. According to the COM-B model a behaviour-change occurs when capability and opportunity are present and when the person is more motivated to enact that behaviour than any other [9]. The existence of a behavioural immune system in humans to control epidemic or pandemic is evident in literature [10]. For example, behaviour change during A/H1N1 influenza pandemic in 2009 [11, 12] and during Zika outbreak [13] reduced the transmission of virus. Following the evidences of behaviour theories and public health principles of infectious disease control, several measurement methods have been adopted and practices globally to control COVID-19 [14–17]. Globally, countries have adopted various controlling measures to fight against the spread of the virus, notably, frequent and proper handwashing, maintaining social distance, imposing lockdown to limit mobility of people, ensuring isolation or quarantine of the infected and suspected persons [5, 18, 19]. Studies showed that knowledge about COVID-19 and prevention measures to be taken, and attitudinal dispositions are significantly associated with appropriate infection prevention practice, which has the potential to play a significant role in the prevention and control of community transmission of COVID-19 [20]. The government of Bangladesh, within a few days of identifying the COVID-19 case, declared a nationwide lockdown for a few months and recommended its people to maintain the standard guidelines to prevent COVID-19 [21–23].

Around 38% (62.5 million) of the total population of this country live in urban areas and about half of them live in slums [24, 25]. In Dhaka City, the capital of Bangladesh, there are more than 5,000 slums which are densely inhabited by an estimated four million people [26]. Almost three-fourths of the slum families live in a single room with shared toilets, bathrooms, shared water sources, poor access to electricity and communication system [27–29]. In addition, they are mainly involved in informal professions with limited earnings. Due to the lack of education, economic vulnerabilities, and living conditions, this population is highly susceptible to COVID-19 infection. The evidence of studies showed that the sociodemographic and economic characteristics of individuals are significantly associated with the level of KAP

against the spread of COVID-19 [30–32]. Although some assessments on KAP towards COVID-19 have been conducted in Bangladesh [20, 30, 31, 33], studies focusing on the urban slum dwellers are very limited [30]. Therefore, rapid assessment of the KAP of slum population towards COVID-19 transmission-prevention is essential in, filling the existing knowledge gap and informing policymakers to design and implement required timely public health interventions in slum areas. Thus, the current study aims to assess the KAP of the slum dwellers towards COVID-19 transmission-prevention.

## Material and methods

### Ethics statement

This study was approved by the Research Review Committee and the Ethics Review Committee of International Centre for Diarrhoeal Disease Research, Bangladesh (icddr,b) (Protocol# PR-20092). Informed verbal consent was obtained from all the respondents before their interview and audio calls were recorded with permission from respondents. The respondents were assured about confidentiality and anonymity of their information.

### Study design

This study was a component of a rapid assessment of the health system impact of COVID-19 among the urban slum population in Bangladesh [34]. This was a cross-sectional telephone-based exploratory study conducted among 476 adult individuals (>18 years) of 476 households from the selected slums during 31 October to 1 December 2020.

### Study setting and population

The study was based on the sampling frame of the existing Urban Health and Demographic Surveillance System (UHDSS) of icddr,b that covers 31,577 households in five slums of the Dhaka North City Corporation, Dhaka South City Corporations, and Gazipur City Corporation. Adult male or female household members constituted the study population.

### Sample design and procedure

We assumed that 50% of the adult slum population have proper KAP towards COVID-19. Considering this proportion (p = 0.5), with 95% confidence level and 5% precision level, an estimated 384 consenting individuals/households were required to be enrolled in the study. Assuming a 10% non-response rate and 1.2 design effect for selecting respondents from different slums, 512 individuals/households were selected by multi-stage sampling technique. To capture the level of KAP, an adult informed member was selected from each household as respondent.

### Data collection instruments and measures

The study instrument was developed by the study team in consultation with experts from different fields to check its relevance and make necessary changes according to the study requirement. The development of instrument was grounded on theory of how knowledge and attitudes of individuals influence healthy behaviour and motivate them in taking actions towards prevention of infection [35–37]. We also reviewed published articles related to the assessment of KAP towards COVID-19 in similar settings, and the general guidelines recommended by the Bangladesh government and WHO [5, 38–40].

Initially, the tools were developed in English and translated into Bengali, contextualized for urban slum people of Bangladesh, pretested to ensure that the respondents understood all the

KAP questions / items for control and transmission prevention of COVID-19, and then finalized the questionnaire for interview [41–49]. Both the initial and translated version of the questionnaire were rigorously reviewed by the Institutional Review Board (IRB) of icddr,b. The questionnaire included several parts, including sociodemographic condition, access to COVID-19 related information, and KAP towards COVID-19 preventive and curative measures.

There were 31 knowledge questions in five sub-domains, namely, about the disease (5), prevention (10), risk groups (6), symptoms (7), and access to COVID-19 related information (3); 10 questions to understand attitude and 16 questions to evaluate prevention practices. There was a total of 57 questions related to KAP and the overall Cronbach α of the questions regarding the final data was 0.86, which indicates a satisfactory level of internal consistency of the items [50]. The knowledge and attitude questions had three levels with "Yes", "No", and "Don't know"; whereas the preventive practice was only responded as "Yes" or "No". Information on KAP level was assessed using the scoring method. Each positive response for knowledge and attitude was scored as "1" and a negative response was scored as "0" [51, 52].

## Data collection

We collected data by direct phone calls to the mobile number of the selected household as it was not feasible to conduct a face-to-face survey during the pandemic due to the risk of infection and restricted mobility. The UHDSS surveillance workers, during their routine data collection, took verbal consent from the households whether they would be interested to take part in this study or not and whether they would like to share their mobile numbers or not. Only the households those agreed to share their mobile phone numbers to take part in this study were used as a sampling frame for this survey. Six trained data collectors conducted interviews with the respondents over the phone. Respondents received BDT 200 (US$2.37) using a mobile financial service to compensate their time for participating in the study.

## Data analysis

Both descriptive and inferential statistical analyses were performed with the Two-tailed t-test for means of variables of two independent samples and the One-way ANOVA for variables of more than two independent samples to test the hypothesis of a statistical difference for each dimension of KAP scores with demographic and socioeconomic categories, separately. Statistical relationship between 'Knowledge-Attitude', 'Knowledge-Practice', and 'Attitude-Practice' were assessed using the pearson's test of correlation. Principal component analysis was used to generate asset scores based on the household's ownership of land and holdings of durable assets e.g., Television, Refrigerator, Mobile phone [53]. Then the asset scores were divided into five quintiles where the first quintile referred to the poorest households and the fifth quintile referred to the richest households. As the number of questions / items were not equal in three domains of the KAP, we developed weighted scores for each of the domains on the scale of 100. We used quartiles for categorization of the weighted scores where the first quartile represented 'poor', the second and third quartiles represented 'average/moderate' while the fourth quartile represented 'good' levels. However, for categorizing the sub-domains of knowledge, we used the score range for overall knowledge for three categories (good, moderate, poor) derived using the quartiles. The attitude score was standardized using z-score and using this z-score we estimated the proportion of individuals with positive and negative attitude.

Three multiple linear regression models were applied to identify the socioeconomic determinants of KAP scores, separately. In these models, KAP level were treated as dependent variables and household headship status, age, gender, education, occupation, marital status,

household size, regular earning status, and wealth status were treated as explanatory variables. The multiple linear regression models were specified as follows:

$$Y_i = \beta_0 + \beta_1 X_{1i} + \beta_2 X_{2i} + \beta_3 X_{3i} + \cdots + \varepsilon_i \ldots \ldots \ldots \ldots \tag{1}$$

Where, $Y_i$ is the knowledge, attitude, or practice scores of i-th individual; $\beta_0$ is a constant; $X_1, X_2, X_3, \ldots$ denote control variables e.g., age, gender, marital status, family size, earning status, asset quintiles; $\beta_1, \beta_2, \beta_3, \ldots$ represent the coefficients, and $\varepsilon_i$ is the random error term of the model. We analyzed the data using STATA version 16 [54].

## Results

### Demographic and socioeconomic characteristics

The results in the study showed that a total of 476 respondents from 476 households who participated were heads (65%) of households with 82% being adults (18–45 years of age), 18% were older than 45 years. The proportion of male respondents was higher (58%) than the female respondents. Regarding the educational qualification, 30% of the respondents had a primary level of education, 28% had a secondary level education, 9% had higher secondary and above level education, and 26% had no education. More than 50% of the respondents did not have regular income and about 54% did not work in the last 30 days before the survey (see Table 1).

### Response to the items for KAP towards COVID-19

Out of the five sub-domains of the knowledge domain, in the knowledge about disease sub-domain, 99% of the respondents reported that they heard about COVID-19 and only 70% reported that COVID-19 was a curable disease. Under the prevention sub-domain, 94% of the respondents reported that cleaning office, home and regularly used equipment with Sanitizer / soap / detergent can reduce the risk of COVID-19. Approximately 65% of the respondents correctly responded that the COVID-19 patient recovers within 28 days. A majority of the respondents (87%) knew that elderly people are more at risk of COVID-19 and 62% believed that children are at risk of COVID-19. Regarding symptoms related knowledge, the highest percentage of respondents reported fever (83%) and the lowest percentage of respondents reported odour-lessness (46%) as the symptoms of COVID-19. About 51% of the respondents mentioned that they received messages about COVID-19 prevention and treatment, and 20% knew that they had access to a quarantine facility nearby their home. In the attitude domain, a majority of the respondents reported that they were aware that an infected person can increase the risk of spreading COVID-19 by roaming freely (96%), infection can be prevented through proper precautions (95%), practicing hygiene (94%), raising awareness (88%), and staying at home (86%). More than half of the respondents incorrectly answered that bathing in hot water could prevent the disease (64%) and that "COVID-19 is the curse of Allah" (66%), while 46% believed that death was the fate of COVID-19.

For the items related to practice, a majority of the respondents reported that they washed their hands in the last 24 hours with soap or sanitizer (98%), wore a mask (94%), maintained social distance (78%) when went outside, and regularly washed their hands for 20–30 seconds (80%). About 80% said they did not eat half-boiled foods, did not spend time with friends (75%), and did not touch mobile phone before washing hands (73%). About 61% of respondents mentioned that they shared bathrooms and toilets with other families and 50% shared food/ water pot (see Table 2).

**Table 1. Characteristics of the respondents and households (N = 476).**

| Characteristics | n | % |
|---|---|---|
| **Relationship with household head** | | |
| Household head | 309 | 64.9 |
| Others | 167 | 35.1 |
| **Age in years** | | |
| ≤30 | 188 | 39.5 |
| 31–45 | 204 | 42.9 |
| 46–60 | 67 | 14.1 |
| >60 | 17 | 3.6 |
| **Sex** | | |
| Male | 277 | 58.2 |
| Female | 199 | 41.8 |
| **Education Level** | | |
| No education | 125 | 26.3 |
| Primary | 145 | 30.5 |
| Secondary and above | 206 | 43.3 |
| **Occupation** | | |
| Housewife | 110 | 23.1 |
| Driver | 72 | 15.1 |
| Service | 70 | 14.7 |
| Business | 63 | 13.2 |
| Day labor | 40 | 8.4 |
| Unemployed/students | 32 | 6.7 |
| retired person | 27 | 5.7 |
| Garment's worker | 22 | 4.6 |
| Technical labor/ Electrician | 20 | 4.2 |
| Housemaid | 13 | 2.7 |
| Others (e.g., beggar) | 7 | 1.5 |
| **Marital Status** | | |
| Married | 405 | 85.1 |
| Unmarried | 53 | 11.1 |
| Others | 18 | 3.8 |
| **Family size** | | |
| < = 3 members | 124 | 26.1 |
| > = 4 to < = 6 members | 314 | 66.0 |
| >6 members | 38 | 8.0 |
| **Regular earning person** | | |
| Yes | 222 | 46.6 |
| No | 254 | 53.4 |
| **Worked in last 30 days** | | |
| No working days | 255 | 53.6 |
| ≤ 10 | 14 | 2.9 |
| 11–20 | 36 | 7.6 |
| > 20 | 171 | 35.9 |
| **Income in last 30 days in BDT (respondents)** | | |
| No income | 259 | 54.4 |
| < = 8000 | 67 | 14.1 |
| >8000 to < = 14000 | 86 | 18.1 |

(*Continued*)

**Table 1.**  (Continued)

| Characteristics | n | % |
|---|---|---|
| >1400 to < = 20000 | 49 | 10.3 |
| >20000 | 15 | 3.2 |
| Asset quintiles | | |
| Poorest | 126 | 26.5 |
| 2nd | 66 | 13.9 |
| 3rd | 95 | 20.0 |
| 4th | 94 | 19.8 |
| Richest | 95 | 20.0 |

## Average weighted scores for KAP and sources of knowledge about COVID-19

Average knowledge score was 67.1 out of 100 with a standard deviation of 15.9. Overall, about 25.2% of the respondents had good knowledge (fourth quartile) towards COVID-19, 49.8% had moderate knowledge (second and third quartile) and the rest had poor knowledge (first quartile). Considering the sub-domains of knowledge, about 80% of the respondents had good knowledge about the COVID-19 disease, about 70% about prevention, 27% about risk group, 45% about symptoms, and 7% about access to information. The average attitude score was 73.6 out of 100 with a standard deviation of 14.4. Overall, 48% of the respondents had positive attitude, 52% had negative attitude towards COVID-19. The average practice score was 65.1 out of 100 with a standard deviation of 16.6. Overall, about 21% of the respondents maintained good practice, followed by moderate practice (46%), and poor practice (33%) towards COVID-19 (see Table 3).

We found that television was the main source for three-quarters of the respondents (75%) to get knowledge about COVID-19 followed by neighbours (6.5%) and social media (5.5%) (Fig 1).

## Correlation among KAP scores

From the Pearson's correlation coefficient test between knowledge-attitude, knowledge-practice and attitude-practice, we found that there was a statistically significant positive correlation of knowledge with attitude, knowledge with practice, and attitude with practice (see Table 4).

## KAP scores across demographic and socioeconomic characteristics

Overall, KAP scores were higher if the respondents were members of a household compared to the respondents who were the head of a household. The attitude and practice score were significantly different by the status of the household head. On average, female respondents had a higher KAP score compared to the males, and there was a statistically significant difference in attitude and practice score between male and female respondents. We observed that KAP scores towards COVID-19 increased with the level of education. Differences in knowledge and attitude scores by educational level were statistically significant. KAP score also significantly varied across the types of occupation. A higher knowledge and attitude score were found among the unmarried respondents, in contrast, a higher practice score was found among the married respondents and the difference in knowledge by marital status was statistically significant. The average of KAP scores were higher among the non-earning respondents compared to the earning respondents and the difference between them was statistically significant.

**Table 2. Knowledge, attitude, and practice towards COVID-19 among the respondents.**

| SL | Questions | Yes | No | Don't know |
|----|-----------|-----|-----|-----|
| | | n (%) | n (%) | n (%) |
| **Knowledge towards COVOD-19** | | | | |
| | **Knowledge about disease** | | | |
| 1 | Have you heard about COVID-19? | 472 (99.16) | 4(0.84) | 0 |
| 2 | Is COVID-19 a contagious disease? | 356 (74.79) | 88 (18.49) | 32(6.72) |
| 3 | Is COVID-19 a viral disease? | 416 (87.39) | 29(6.09) | 31(6.51) |
| 4 | COVID-19 is a curable disease? | 333 (69.96) | 90 (18.91) | 53(11.13) |
| 5 | COVID-19 is a deadly disease? | 417 (87.61) | 42(8.82) | 17(3.57) |
| | **Knowledge about prevention** | | | |
| 6 | Does COVID-19 transmit through air? | 349 (73.32) | 74 (15.55) | 53(11.13) |
| 7 | Do you know that symptoms of COVID-19 become visible within 3–14 days of infection? | 352 (73.95) | 69(14.5) | 55(11.55) |
| 8 | Do you know that generally a COVID-19 patient gets recovered within 28 days? | 311 (65.34) | 61 (12.82) | 104 (21.85) |
| 9 | Can a person be infected with COVID-19 after recovered from it? | 316 (66.39) | 70 (14.71) | 90(18.91) |
| 10 | Treatment of COVID-19 can be done at home? | 350 (73.53) | 89(18.7) | 37(7.77) |
| 11 | Can COVID-19 be prevented through using face mask? | 381 (80.04) | 73 (15.34) | 22(4.62) |
| 12 | Without proper disinfecting, usage of cloth mask can increase the risk of infection? | 440 (92.44) | 25(5.25) | 11(2.31) |
| 12 | Can frequent hand washing prevent COVID-19? | 416 (87.39) | 46(9.66) | 14(2.94) |
| 13 | Maintaining 1 meter or 3 feet distance from infected person is helpful for preventing COVID-19? | 412 (86.55) | 47(9.87) | 17(3.57) |
| 14 | Cleaning office, home and regular usage things with Sanitizer/soap/detergent can reduce the risk of COVID-19? | 448 (94.12) | 18(3.78) | 10(2.1) |
| | **Knowledge about high-risk group** | | | |
| 15 | Elderly people are more at risk of COVID-19? | 416 (87.39) | 39(8.19) | 21(4.41) |
| 16 | Smokers are at risk of COVID-19? | 372 (78.15) | 54 (11.34) | 50(10.5) |
| 17 | Rich people are at risk of COVID-19? | 302 (63.45) | 111 (23.32) | 63(13.24) |
| 18 | Children are at risk of COVID-19? | 297 (62.39) | 148 (31.09) | 31(6.51) |
| 19 | Pregnant women are at risk of COVID-19? | 305 (64.08) | 100 (21.01) | 71(14.92) |
| 20 | COVID-19 can be serious to the patients with diabetes and heart disease? | 403 (84.66) | 33(6.93) | 40(8.4) |
| | **Knowledge about symptoms** | | | |
| 21 | Whether the followings are symptoms of COVID-19 | | | |
| 22 | Fever | 394 (82.77) | 59 (12.39) | 23(4.83) |

(*Continued*)

**Table 2.** (Continued)

| SL | Questions | Yes | No | Don't know |
|---|---|---|---|---|
| | | n (%) | n (%) | n (%) |
| 23 | Dry cough | 376 (78.99) | 62 (13.03) | 38(7.98) |
| 24 | Difficulty in breathing | 390 (81.93) | 57 (11.97) | 29(6.09) |
| 25 | Sore throat | 384 (80.67) | 59 (12.39) | 33(6.93) |
| 26 | Odor lessness | 218 (45.8) | 122 (25.63) | 136 (28.57) |
| 27 | Diarrhea | 236 (49.58) | 154 (32.35) | 86(18.07) |
| 28 | Weakness | 247 (51.89) | 143 (30.04) | 86(18.07) |
| | **Knowledge about access to information** | | | |
| 29 | Do you have access to quarantine facility nearby your house/workplace? | 94 (19.75) | 347 (72.9) | 35(7.35) |
| 30 | Did you receive any message on COVID-19 prevention and treatment? | 244 (51.26) | 232 (48.74) | 0 |
| 31 | Do you know any hotline number to contact in case of any symptoms of you or others? | 234 (49.16) | 242 (50.84) | 0 |
| **Attitudes towards COVID-19** | | | | |
| 1 | Do you think, COVID-19 can be prevented? | 419 (88.03) | 24(5.04) | 33(6.93) |
| 2 | Does bathing in hot water reduce risk of COVID-19? | 305 (64.08) | 60 (12.61) | 111 (23.32) |
| 3 | Do you think that dwelling place, sitting place, crowded place or public transport handle (bus, train, taxi, auto rickshaw or rickshaw) are the major sources of COVID-19 infection? | 420 (88.24) | 28(5.88) | 28(5.88) |
| 4 | Staying at home, COVID-19 can be prevented | 409 (85.92) | 41(8.61) | 26(5.46) |
| 5 | Infected person can increase the risk of spreading COVID-19 by roaming freely? | 456 (95.8) | 8(1.68) | 12(2.52) |
| 6 | Practicing hygiene can reduce the risk of COVID-19? | 446 (93.7) | 19(3.99) | 11(2.31) |
| 7 | COVID-19 can be prevented through proper precautions? | 453 (95.17) | 17(3.57) | 6(1.26) |
| 8 | Awareness is sufficient for COVID-19 prevention? | 420 (88.24) | 39(8.19) | 17(3.57) |
| 9 | COVID-19 is the curse of ALLAH | 313 (65.76) | 93 (19.54) | 70(14.71) |
| 10 | Death is the fate of COVID-19? | 220 (46.22) | 215 (45.17) | 41(8.61) |
| **Practice towards COVID-19** | | | | |
| 1 | Do you sneeze into your elbow? | 289 (60.71) | 187 (39.29) | - |
| 2 | Do you often touch your mouth, eye, and nose? | 259 (54.41) | 217 (45.59) | - |
| 3 | Do you often eat half boiled fish, meat, egg, or vegetables? | 96 (20.17) | 380 (79.83) | - |
| 4 | Do you clean your house with detergent/disinfectant regularly? | 372 (78.15) | 104 (21.85) | - |

(*Continued*)

**Table 2.** (Continued)

| SL | Questions | Yes | No | Don't know |
|----|-----------|-----|-----|------------|
|    |           | n (%) | n (%) | n (%) |
| 5 | Do you disinfect your mobile phone with sanitizer regularly? | 256 (53.78) | 220 (46.22) | - |
| 6 | Do you drink tea/coffee at street stall? | 193 (40.55) | 283 (59.45) | - |
| 7 | Do you usually spend time with your friends? | 118 (24.79) | 358 (75.21) | - |
| 8 | Do you touch your clothes, money bag, key ring, plate, glass, laptop, earphone, mobile before washing your hands? | 128 (26.89) | 348 (73.11) | - |
| 9 | Do you use public transport? | 295 (61.97) | 181 (38.03) | - |
| 10 | Do you maintain social distance when you go outside? | 369 (77.52) | 107 (22.48) | - |
| 11 | Do you wash your hands for 20–30 second regularly? | 381 (80.04) | 95 (19.96) | - |
| 12 | In the last 24 hours, did you wash your hand using soap or sanitizer? | 468 (98.32) | 8(1.68) | - |
| 13 | Do you wear mask when you go outside? | 448 (94.12) | 28(5.88) | - |
| 14 | Do you share food/water pot with other family members? | 237 (49.79) | 239 (50.21) | - |
| 15 | Do you share bathroom with other families? | 290 (60.92) | 186 (39.08) | - |
| 16 | Do you share toilets with other families? | 290 (60.92) | 186 (39.08) | - |

Similarly, the average KAP scores were higher among the person who did not work or worked up to 10 days in the last 30 days and the differences across the categories in knowledge and attitude score were statistically significant. We found that both knowledge and practice scores were higher among respondents who had an income of more than BDT 20,000 in the last 30 days. However, the attitude score was higher for the respondents who did not have an income. There was a statistically significant difference in attitude score among the respondents across the income group. Furthermore, we found that the respondents from richest quintile had higher KAP scores and the variation of scores of all three domains were statistically significant across the asset quintiles (see Table 5).

## Determinants of KAP towards COVID-19

The crude and adjusted association of the dependent variables i.e., knowledge, attitude, and practice scores with independent socioeconomic characteristics of the respondents i.e., age, sex, education level, occupation, asset quantiles, etc. are presented in Table 6. In the crude association, we found that age, education level, occupation, marital status, and asset quintiles were significantly associated with the knowledge score. However, in the adjusted association, compared to the group without education, the knowledge score was significantly higher among the respondents who had primary ($\beta$ = 8.554; p<0.001) and secondary and above ($\beta$ = 11.431; p<0.001) education level. The knowledge score was significantly higher among the respondents from the fourth ($\beta$ = 4.463; p<0.05) and the richest ($\beta$ = 3.585; p<0.001) quintiles compared to the respondents from the poorest quintile.

**Table 3. Average level of knowledge, attitude, and practice towards COVID-19 among the respondents.**

| Variable | No of questions | Weighted scale | Weighted mean ± SD | 95% CI | Level (%) | | |
|---|---|---|---|---|---|---|---|
| | | | | | Good | Moderate | Poor |
| Overall knowledge domain | 31 | 100 | 67.1 ± 15.9 | (65.6–68.5) | 25.2 | 49.8 | 25.0 |
| Knowledge about disease | 5 | 20 | 16.8 ± 4.0 | (16.4–17.1) | 79.8 | 13.5 | 6.7 |
| Knowledge about prevention | 10 | 20 | 15.9 ± 4.1 | (15.5–16.2) | 69.8 | 18.7 | 11.6 |
| Knowledge about risk group | 6 | 20 | 12.9 ± 3.4 | (12.6–13.3) | 26.9 | 43.1 | 30.0 |
| Knowledge about symptoms | 7 | 20 | 13.5 ± 6.5 | (12.9–14.1) | 45.2 | 17.0 | 37.8 |
| Knowledge about access to information | 3 | 20 | 8.0 ± 5.9 | (7.5–8.5) | 6.7 | 31.5 | 61.8 |
| Practice domain | 16 | 100 | 65.1 ± 16.6 | (63.6–66.6) | 21.2 | 46.2 | 32.6 |
| | | | | | Positive | Negative | |
| Attitude domain | 10 | 100 | 73.6 ± 14.4 | (72.3–74.9) | 47.7 | 52.3 | |

We found that attitude score was significantly decreasing with the increase in respondents' age. Attitude score was significantly higher among the respondents who had primary level (β = 4.563; p<0.01) and secondary and above level of education (β = 4.204; p<0.05) compared to the respondents having no-education. In terms of occupation, the attitude score was significantly lower among the day labourer compared to the housewives. Furthermore, attitude score was significantly higher among the respondents from the richest quintile (β = 4.066, p<0.05) compared to the respondents from poorest quintile.

While examining the association of practice score with the characteristics of the participants, we found that the female respondents had significantly higher practice scores compared to the male counterparts (β = 6.841; p<0.01). Divorced/ widowed respondents had a significantly lower practice score compared to those married (β = -10.579; p<0.05). The practice score had a positive association with the asset quintile; respondents from the richest quintile had significantly higher practice score by 6.5 points compared to the respondents from the poorest quintile.

## Discussion

The study sought to assess the KAP of selected slum dwellers in Dhaka city, Bangladesh towards COVID-19 and found that a small proportion of the sample surveyed (25.2%) demonstrated good knowledge regarding the infection, almost half of them (48%) expressed a positive attitude about controlling and preventing this disease, and like good knowledge level, a small

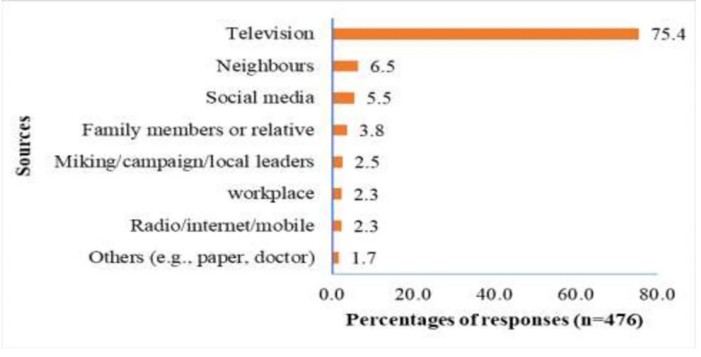

**Fig 1. Sources of information on COVID-19 (in percentage).**

**Table 4. Correlation between knowledge, attitude, and practice scores.**

| Variable | Correlation coefficient | p-value* |
|---|---|---|
| Knowledge-Attitude | 0.4181 | **0.000** |
| Knowledge-Practice | 0.3122 | **0.000** |
| Attitude-Practice | 0.1226 | **0.007** |

*Pearson's correlation

proportion of the sample surveyed (21%) had favourable practices towards COVID-19 transmission control and prevention.

Slum dwellers in Bangladesh reside in congested spaces, while having a very poor surrounding environment, thus, these people are more susceptible to COVID-19 infection [55, 56]. A cross-sectional study in Bangladesh reported that the seroprevalence rate of COVID-19 was the highest (64%) in slum areas, whereas 38% in urban areas and 29% in rural areas [57]. The current assessment, therefore, is useful to inform the policymakers about the level and the dimensions of KAP and the associated determinants to design targeted interventions.

A recent study conducted in slum areas reported that the overall correct rate of knowledge of COVID-19 was 36%, which was lower compared to our study (67%). However, the correct attitude and practice rate reported in that study was higher compared to the current study (88% vs 74%) and (82% vs 65%), respectively [30]. The possible reason for this difference could be the difference in data collection period and choices of items for the different dimensions of KAP. Ferdous et al. (2020) conducted a study on general population that reported 48.3% of the respondents had more accurate knowledge, 62.3% had positive attitude, and 55.2% had good practice. Compared to the general population, lower proportion of slum population had good knowledge (48.3% vs 25.2%), good attitude level (62.3% vs 48%) and good practice (55.2% vs 21%) [58]. As mentioned earlier that the slum people live in a disadvantaged environment have lower access to information which was also reflected in the findings of our study (only 6.7% had good access to information). Moreover, they live in a congested place for which they might have limited scope of maintaining good practice towards COVID-19 without having a strong motivation. Another study on KAP towards COVID-19 conducted among Bangladeshi youth reported that a higher proportion of the respondents had good knowledge (61.2% vs 25.2%), positive attitude (79% vs 48%), and good practices (52% vs 21%) compared to our study [59]. In our study, it was also evident that KAP score was higher among the youth respondents compared to the elderly. It is noteworthy that in all three studies mentioned above, the reported attitude score was higher compared to the knowledge and practice score, which is similar to our findings. This might be attributable to the fact that the attitude reflects the respondents believe and perception which is different from their practice and the existing level of knowledge.

Our study also revealed that 75% of the slum people depended on television to gather COVID-19 related information, which is remarkably higher than other sources, for example, internet, social media. This finding is similar to a study that was conducted to assess the KAP on COVID-19 in the slum area of Dhaka, Bangladesh [30]. Another study conducted to evaluate the KAP in all the administrative districts of Bangladesh also reported a similar finding on people's access to information [60]. The lack of appropriate campaign, limited internet access, smartphone possession, and low access to health information could be the reasons for having such lower proportion of slum dwellers with good knowledge about the COVID-19. Therefore, interventions such as health education programs incorporating mass media with a variety of advertisements can be influential in the dissemination of information about COVID-19.

**Table 5. Average level of knowledge, attitude, and practice scores towards COVID-19 by the demographic and socioeconomic characteristics of respondents.**

| Description | n | Knowledge Score (Mean ± SD) | p- value | Attitude Score (Mean ± SD) | p- value | Practice Score (Mean ± SD) | p- value |
|---|---|---|---|---|---|---|---|
| **Characteristics** | | | | | | | |
| **Relationship with HH head** [a)] | | | | | | | |
| Household head | 309 | 66.1± 16.3 | 0.065 | 72.2± 15.4 | 0.005 | 63.9± 16.8 | 0.028 |
| Others | 167 | 68.9± 15.0 | | 76± 11.9 | | 67.4± 16 | |
| **Age group** [b)] | | | | | | | |
| ≤30 | 188 | 68.7± 15.2 | 0.098 | 76.2± 12 | 0.008 | 65± 15.7 | 0.348 |
| 31–45 | 204 | 66.7± 15.4 | | 72.5± 14.5 | | 66.4± 16.7 | |
| 46–60 | 67 | 65± 16.9 | | 70.4± 17.2 | | 62.5± 17.6 | |
| >60 | 17 | 60.4± 22.4 | | 70± 20.3 | | 62.5± 21.2 | |
| **Sex** [a)] | | | | | | | |
| Male | 277 | 66.5± 16.6 | 0.373 | 72.4± 15.1 | 0.038 | 62.3± 16.3 | 0.038 |
| Female | 199 | 67.8± 14.9 | | 75.2± 13.1 | | 69.1± 16.2 | |
| **Education Level** [b)] | | | | | | | |
| No education | 125 | 59.5± 17.8 | 0.001 | 68.6± 16.4 | 0.001 | 63.3± 17.1 | 0.352 |
| Primary | 145 | 68.1± 14.1 | | 74.6± 13 | | 65.9± 15.9 | |
| Secondary and above | 206 | 70.9± 14.3 | | 75.9± 13.2 | | 65.7± 16.8 | |
| **Occupation** [b)] | | | | | | | |
| Housewife | 70 | 67.5± 19.3 | 0.037 | 74.1± 13 | 0.005 | 65.3± 17.1 | 0.001 |
| Driver | 63 | 67.9± 13.6 | | 74.6± 11.6 | | 68.1± 14.6 | |
| Service | 110 | 67.1± 15 | | 76.2± 11.3 | | 69.4± 16.5 | |
| Business | 13 | 65.2± 8.3 | | 66.9± 14.4 | | 73.1± 10.9 | |
| Unemployed/students | 72 | 63.3± 16.7 | | 68.1± 18.3 | | 58.6± 15.1 | |
| Day labor | 40 | 63.4± 17.3 | | 69.5± 17.4 | | 57.7± 16.9 | |
| Retired person | 20 | 67.0± 15.8 | | 78.0± 13.6 | | 66.6± 19.6 | |
| Technical labor | 27 | 75.3± 10.1 | | 76.7± 10.7 | | 70.4± 15.5 | |
| Garment worker | 22 | 73.0± 8.7 | | 75.9± 11.4 | | 66.2± 14.8 | |
| Housemaid | 32 | 68.8± 18.1 | | 74.7± 18.5 | | 60.2± 18.2 | |
| Others (e.g., beggar) | 7 | 59.0± 19.4 | | 72.9± 12.5 | | 61.6± 9.1 | |
| **Marital Status** [b)] | | | | | | | |
| Married | 405 | 66.3± 15.9 | 0.025 | 73.2± 14.5 | 0.234 | 65.4± 16.3 | 0.388 |
| Unmarried | 53 | 72.6± 14.4 | | 75.1± 10.7 | | 64.5± 16.9 | |
| Others | 18 | 66.9± 17.8 | | 78.3± 20.1 | | 60.1± 22.3 | |
| **Family size** [b)] | | | | | | | |
| < = 3 members | 124 | 66.3± 15.2 | 0.733 | 74.4± 15.2 | 0.768 | 64.5± 17.4 | 0.835 |
| > = 4 to < = 6 members | 314 | 67.2± 16.5 | | 73.2± 14.4 | | 65.3± 16.2 | |
| >6 members | 38 | 68.5± 13.4 | | 73.7± 10.5 | | 66.1± 17.3 | |
| **Regular earning person** [b)] | | | | | | | |
| Yes | 222 | 65.6± 16.4 | 0.054 | 71.4± 15.6 | 0.002 | 63.6± 16.3 | 0.058 |
| No | 254 | 68.4± 15.4 | | 75.4± 12.9 | | 66.5± 16.8 | |
| **Worked in last 30 days** [b)] | | | | | | | |
| No working days | 255 | 68.3± 15.4 | 0.039 | 75.5± 12.9 | 0.001 | 66.3± 17 | 0.354 |
| ≤ 10 | 14 | 68.1± 15.7 | | 77.1± 9.1 | | 60.7± 21.3 | |
| 11 to 20 | 36 | 60.4± 18.1 | | 66.7± 18.2 | | 63.4± 14.1 | |
| > 20 | 171 | 66.5± 15.9 | | 71.9± 15.3 | | 64.1± 16.1 | |
| **Income in last 30 days** [b)] **(BDT)** | | | | | | | |
| No income | 259 | 68.1± 15.4 | 0.150 | 75.3± 13.1 | 0.045 | 66.5± 17 | 0.081 |

*(Continued)*

**Table 5.** (Continued)

| Description | n | Knowledge Score (Mean ± SD) | p- value | Attitude Score (Mean ± SD) | p- value | Practice Score (Mean ± SD) | p- value |
|---|---|---|---|---|---|---|---|
| < = 8000 | 67 | 66.9± 14.7 | | 73.4± 13.7 | | 64± 14.9 | |
| >8000 to < = 14000 | 86 | 63.8± 17.8 | | 71± 15.8 | | 63.8± 16.3 | |
| >14000-< = 20000 | 49 | 65.7± 17 | | 69.8± 17.6 | | 60.2± 15.8 | |
| >20000 | 15 | 72.3± 11.9 | | 72± 15.2 | | 70± 18.3 | |
| Asset quintiles [b] | | | | | | | |
| Poorest | 126 | 63.4± 17.5 | 0.001 | 72.1± 14.7 | 0.008 | 62.2± 17.7 | 0.004 |
| 2nd | 66 | 61.9± 17.6 | | 70.5± 15.1 | | 65.7± 15.6 | |
| 3rd | 95 | 66.2± 15.6 | | 72.2± 14.7 | | 64.7± 17.6 | |
| 4th | 94 | 68.3± 14.3 | | 75.2± 14.3 | | 63.7± 16.2 | |
| Richest | 95 | 75± 10.5 | | 77.5± 12.1 | | 70.5± 14 | |

[a] T-test of independence

[b] One-way ANOVA test

Two important characteristics e.g., education and asset quintile were significantly associated with the knowledge and attitude of the respondents. We found that higher education was significantly associated with having a higher knowledge score towards COVID-19. A similar association of knowledge related to COVID-19 with higher education were observed in other studies in Bangladesh [30] and Iran [39]. Educated individuals might have good access to information and are aware of the potential impact of COVID-19 from a variety of sources for example newspaper. As a result, they have more knowledge on COVID-19 compared to the uneducated individuals. This finding indicated the importance of educational intervention to improve knowledge of COVID-19 among the slum dwellers. Previous studies have also reported that educational interventions had a significant impact on improving knowledge level [30, 58, 61]. We found that knowledge and attitude were significantly correlated. Comparatively higher educated respondents also had a positive attitude towards COVID-19. A similar association of attitude with education level was observed in other recent local and international studies [30, 58, 62].

Among slum people, those who belonged to the richer quintile had a significantly higher level of KAP compared to the poorest quintile. The economic ability of respondents enables them to access information about COVID-19 which might be a significant factor behind this phenomenon.

About half of the respondents (50%) believed that death is the ultimate fate of COVID-19. This may indicate the limited access to accurate and timely information among slum population [60]. Only one in five persons maintained good practice against the COVID-19 spreading in slum areas. Another KAP study in Bangladesh revealed a similar result that most of the respondents had poor practice towards COVID-19 [59]. The higher prevalence of poor practice may be the result of existing dwelling conditions in the slum areas. About 60% of the respondents stated that they shared their toilets and bathrooms with other households. Furthermore, due to their socioeconomic status, they had to move outside frequently for earning their livelihoods. We found that female respondents had a significantly higher practice score compared to the male. In the context of Bangladesh, the male is the main wage earner in a household. Therefore, they have to go outside for work and travel by local transport, which ultimately exposes them to COVID-19.

**Table 6. Socioeconomic determinants of knowledge, attitude, and practice towards COVOD-19.**

| Characteristics | n | Model 1: Knowledge | | Model 2: Attitude | | Model 3: Practice | |
|---|---|---|---|---|---|---|---|
| | | Crude coef. (95% CI) | Adjusted coef. (95% CI) | Crude coef. (95% CI) | Adjusted coef. (95% CI) | Crude coef. (95% CI) | Adjusted coef. (95% CI) |
| **Household head status** | | | | | | | |
| Head | 309 | Ref. | Ref. | Ref. | Ref. | Ref. | Ref. |
| Members | 167 | 2.821 (-0.17,5.82) | -0.659 (-5.29,3.97) | 3.815** (1.12,6.51) | 1.362 (-2.91,5.64) | 3.487* (0.37,6.61) | -2.992 (-7.96,1.97) |
| **Age group** | | | | | | | |
| ≤30 | 188 | Ref. | Ref. | Ref. | Ref. | Ref. | Ref. |
| 31–45 | 204 | -2.005 (-5.15,1.14) | -1.281 (-4.88,2.31) | -3.670* (-6.50,-0.84) | -4.984** (-8.30,-1.67) | 1.431 (-1.87,4.73) | 2.970 (-0.88,6.82) |
| 46–60 | 67 | -3.710 (-8.14,0.72) | -1.941 (-6.93,3.05) | -5.722** (-9.70,-1.74) | -6.822** (-11.42,-2.22) | -2.460 (-7.10,2.18) | -0.285 (-5.63,5.06) |
| >60 | 17 | -8.371* (-16.26,-0.48) | -7.947 (-16.42,0.52) | -6.170 (-13.25,0.91) | -9.409* (-17.23,-1.59) | -2.460 (-10.72,5.80) | 0.485 (-8.59,9.56) |
| **Sex** | | | | | | | |
| Male | 277 | Ref. | Ref. | Ref. | Ref. | Ref. | Ref. |
| Female | 199 | 1.317 (-1.59,4.22) | 0.349 (-4.37,5.07) | 2.757* (0.14,5.37) | -0.842 (-5.20,3.51) | 6.853*** (3.88,9.82) | 6.841** (1.78,11.90) |
| **Education Level** | | | | | | | |
| No education | 125 | Ref. | Ref. | Ref. | Ref. | Ref. | Ref. |
| Primary | 145 | 8.554*** (4.90,12.21) | 7.611*** (3.89,11.33) | 5.992*** (2.62,9.36) | 4.562** (1.13,8.00) | 2.605 (-1.38,6.59) | 1.707 (-2.28,5.69) |
| Secondary and above | 206 | 11.431*** (8.04,14.82) | 8.903*** (5.12,12.69) | 7.362*** (4.23,10.49) | 4.204* (0.71,7.70) | 2.416 (-1.28,6.12) | 0.019 (-4.04,4.08) |
| **Occupation** | | | | | | | |
| Housewife | 110 | Ref. | Ref. | Ref. | Ref. | Ref. | Ref. |
| Driver | 72 | 0.376 (-5.00,5.75) | -0.365 (-5.75,5.02) | 0.460 (-4.36,5.28) | 0.381 (-4.59,5.35) | 2.788 (-2.71,8.29) | 3.046 (-2.72,8.82) |
| Service | 70 | -0.456 (-5.19,4.28) | -5.064 (-11.68,1.56) | 2.039 (-2.21,6.28) | -4.345 (-10.45,1.77) | 4.107 (-0.73,8.95) | 0.119 (-6.98,7.21) |
| Business | 63 | -2.344 (-11.69,7.00) | 1.159 (-8.52,10.84) | -7.220 (-15.61,1.17) | -5.999 (-14.93,2.93) | 7.809 (-1.75,17.37) | 5.206 (-5.17,15.58) |
| Day labor | 40 | -4.279 (-9.47,0.92) | -2.676 (-8.04,2.68) | -6.087* (-10.75,-1.43) | -6.111* (-11.06,-1.16) | -6.674* (-11.99,-1.36) | -4.854 (-10.60,0.89) |
| Unemployed/students | 32 | -4.147 (-10.28,1.99) | -1.235 (-7.49,5.02) | -4.643 (-10.15,0.86) | -4.434 (-10.20,1.34) | -7.612* (-13.89,-1.34) | -5.877 (-12.58,0.82) |
| Retired person | 27 | -0.554 (-8.40,7.29) | -2.225 (-10.09,5.64) | 3.857 (-3.18,10.90) | 2.423 (-4.83,9.68) | 1.295 (-6.73,9.32) | 2.281 (-6.14,10.71) |
| Garment workers | 22 | 7.708* (0.70,14.72) | -2.159 (-11.34,7.02) | 2.524 (-3.77,8.81) | -4.423 (-12.89,4.05) | 5.103 (-2.07,12.27) | 5.465 (-4.37,15.30) |
| Technical labor | 20 | 5.433 (-2.13,13.00) | 6.786 (-0.54,14.11) | 1.766 (-5.02,8.55) | 1.240 (-5.52,8.00) | 0.925 (-6.81,8.66) | 2.172 (-5.68,10.02) |
| Housemaid | 13 | 1.296 (-5.31,7.90) | -3.141 (-10.42,4.13) | 0.545 (-5.38,6.47) | -3.100 (-9.81,3.61) | -5.112 (-11.87,1.64) | -5.732 (-13.53,2.07) |
| Others (e.g., beggar) | 7 | -8.559 (-20.83,3.71) | -4.202 (-16.25,7.84) | -1.286 (-12.29,9.72) | 1.006 (-10.11,12.12) | -3.661 (-16.21,8.89) | -5.152 (-18.06,7.76) |
| **Marital status** | | | | | | | |
| Married | 405 | Ref. | Ref. | Ref. | Ref. | Ref. | Ref. |
| Unmarried | 53 | 6.268** (1.73,10.81) | 1.074 (-5.12,7.27) | 1.934 (-2.18,6.05) | -3.828 (-9.55,1.89) | -0.943 (-5.71,3.82) | -1.372 (-8.01,5.27) |
| Others (divorce/widow) | 18 | 0.610 (-6.88,8.10) | 3.610 (-4.16,11.38) | 5.173 (-1.62,11.96) | 9.326* (2.16,16.49) | -5.378 (-13.24,2.48) | -10.579* (-18.90,-2.26) |
| **Family size** | | | | | | | |
| < = 3 members | 124 | Ref. | Ref. | Ref. | Ref. | Ref. | Ref. |
| > 4 to < = 6 members | 214 | 0.866 (-2.45,4.19) | 1.874 (-1.48,5.23) | -1.106 (-4.10,1.89) | 0.079 (-3.02,3.18) | 0.821 (-2.65,4.29) | 0.678 (-2.92,4.28) |
| > 6 members | 38 | 2.231 (-3.57,8.03) | 3.131 (-2.64,8.90) | -0.671 (-5.91,4.57) | 0.416 (-4.91,5.74) | 1.653 (-4.41,7.71) | 1.697 (-4.48,7.88) |
| **Regular earning** | | | | | | | |
| Yes | 222 | Ref. | Ref. | Ref. | Ref. | Ref. | Ref. |
| No | 254 | 2.815 (-0.05,5.68) | 5.007* (1.10,8.91) | 3.992** (1.42,6.56) | 4.895** (1.29,8.50) | 2.888 (-0.10,5.88) | 1.900 (-2.28,6.08) |
| **Asset quintiles** | | | | | | | |

*(Continued)*

**Table 6.** (Continued)

| Characteristics | n | Model 1: Knowledge | | Model 2: Attitude | | Model 3: Practice | |
|---|---|---|---|---|---|---|---|
| | | Crude coef. (95% CI) | Adjusted coef. (95% CI) | Crude coef. (95% CI) | Adjusted coef. (95% CI) | Crude coef. (95% CI) | Adjusted coef. (95% CI) |
| Poorest | 126 | Ref. | Ref. | Ref. | Ref. | Ref. | Ref. |
| Second | 66 | -1.473 (-6.05,3.10) | -1.073 (-5.59,3.44) | -1.609 (-5.85,2.63) | -2.667 (-6.83,1.50) | 3.517 (-1.38,8.42) | 3.236 (-1.60,8.07) |
| Third | 95 | 2.822 (-1.27,6.91) | 2.204 (-1.83,6.24) | 0.147 (-3.65,3.94) | -1.006 (-4.73,2.72) | 2.469 (-1.91,6.85) | 1.012 (-3.31,5.34) |
| Fourth | 94 | 4.860* (0.76,8.96) | 4.436* (0.35,8.52) | 3.149 (-0.66,6.96) | 2.979 (-0.79,6.75) | 1.494 (-2.90,5.89) | 0.676 (-3.70,5.06) |
| Richest | 95 | 11.608*** (7.52,15.70) | 10.217*** (6.05,14.39) | 5.410** (1.62,9.20) | 4.066* (0.22,7.91) | 8.324*** (3.94,12.71) | 6.501** (2.03,10.97) |
| Constant | | | 56.326*** (49.91,62.74) | | 72.916*** (66.99,78.84) | | 59.035*** (52.16,65.91) |
| N | | | 476 | | 476 | | 476 |
| p< | | | 0.000 | | 0.000 | | 0.000 |
| R-squared | | | 0.177 | | 0.140 | | 0.133 |

$^*p<0.05$,

$^{**}p<0.01$,

$^{***}p<0.001$

## Limitations of the study

This study had some important limitations. Firstly, the respondents for this study included only the households with mobile phone users. Data were collected over mobile phone calls, which is a new process in Bangladesh. A majority of the consented respondents may include those who were more concerned about the COVID-19 emergency, which might include response bias. Therefore, the results may not be generalizable to other population who are not mobile phone users. Secondly, this is a cross-sectional study; thus, causal inferences cannot be drawn between the significant socioeconomic characteristics and the KAP level. Thirdly, as the study was a component of a rapid assessment, the questions related to KAP towards COVID-19 transmission control and prevention were adopted and contextualized for Bangladesh from published literature. Health-seeking paradigm of the health belief model could have been applied to validate the study instrument which would have strengthened the study findings [36]. However, this is one of the few studies that assess the KAP of urban slum dwellers in Bangladesh towards COVID-19 transmission prevention.

## Conclusions

This study provides a comprehensive assessment of KAP levels with respect to COVID-19 from the urban slum dwellers in Bangladesh. Overall, of the slum dwellers included in this study, one-fourth demonstrated poor knowledge, almost half had a good attitude level, and one-third maintained poor practice towards COVID-19. However, they had good knowledge about the disease and its prevention but had lack of knowledge about symptoms of the disease, risk population group, and access to COVID-19 related information. Despite one-third of the respondents having poor practice level, a majority of the respondents used a mask, washed their hands for 20–30 seconds, and used soap or sanitizer. Valuable insights on demographic characteristics associated with KAP among the slum population can help policymakers in designing health education programs, awareness raising campaigns, and behavioural change communication interventions. These programs can be designed in consultation with political, religious, and other influential community-based groups. Importance should be given to the

groups who have lower KAP scores, such as individuals aged more than 60 years, uneducated, driver, day labourer, unemployed, people who were regular earning person of a household and belong to the poor socioeconomic group. Expanding living space is not possible in slums; thus, prioritizing household hygiene may enable them to maintain practices to the extent possible. Further studies, with more rigorously developed and validated KAP instruments can be conducted to obtain a more robust estimate of KAP towards COVID-19 transmission-prevention and compare the findings with the current study.

## Supporting information

**S1 Data.**
(XLSX)

## Acknowledgments

icddr,b acknowledges with gratitude the commitment of the Swedish International Development Cooperation Agency to its research efforts and funding for this study. icddr,b is also thankful to the Governments of Bangladesh, Canada, Sweden, and the UK for providing core/unrestricted support.

## Author Contributions

**Conceptualization:** Md. Zahid Hasan, Shehrin Shaila Mahmood.

**Data curation:** Md. Zahid Hasan, Shehrin Shaila Mahmood.

**Formal analysis:** Md. Zahid Hasan, Md. Golam Rabbani, Shehrin Shaila Mahmood.

**Funding acquisition:** Md. Zahid Hasan.

**Investigation:** Md. Zahid Hasan, Shehrin Shaila Mahmood.

**Methodology:** Md. Zahid Hasan, Shehrin Shaila Mahmood.

**Project administration:** Md. Zahid Hasan, Shehrin Shaila Mahmood.

**Software:** Md. Zahid Hasan.

**Supervision:** Shehrin Shaila Mahmood.

**Validation:** Md. Zahid Hasan, Shehrin Shaila Mahmood.

**Writing – original draft:** Md. Zahid Hasan, A. M. Rumayan Hasan, Md. Golam Rabbani, Mohammad Abdus Selim, Shehrin Shaila Mahmood.

**Writing – review & editing:** Md. Zahid Hasan, A. M. Rumayan Hasan, Md. Golam Rabbani, Mohammad Abdus Selim, Shehrin Shaila Mahmood.

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
