## [Decision Letter · Decision Letter 0]

4 Apr 2022

PGPH-D-22-00360

Knowledge, attitude, and practice of Bangladeshi urban slum dwellers towards COVID-19: A cross-sectional study

Dear Dr. Hasan,

Thank you for submitting your manuscript to PLOS Global Public Health. After careful consideration, we feel that it has merit but does not fully meet PLOS Global Public Health’s publication criteria as it currently stands. Therefore, we invite you to submit a revised version of the manuscript that addresses the points raised during the review process.

EDITOR: Kindly consider the following as these have been observed to require attention in the manuscript;

The title requires modifications to reflect what the study aimed to accomplish,The abstract needs revision to reflect the variables and concepts underpinning the study,The data analysis has flaws to be addressed,Revise the data analysis and reflect the research questions to be answered in the discussion.

We look forward to receiving your revised manuscript.

Kind regards,

Nnodimele Onuigbo Atulomah, PhD

Academic Editor

Journal Requirements:

1. Your co-authors:

A M Rumayan Hasan -rumayan@icddrb.org

Md. Golam Rabbani -golam.rabbani2@icddrb.org

Mohammad Abdus Selim -maselim@icddrb.org

Shehrin Shaila Mahmood -shaila@icddrb.org

,have not confirmed authorship of the manuscript. We have resent them the authorship confirmation email; however please check that the above email address for them is correct and follow up personally to ensure they confirm. 

Please note that we cannot proceed your manuscript  until we have received confirmations from all co-authors.

2. Please update the completed 'Competing Interests' statement, including any COIs declared by your co-authors. If you have no competing interests to declare, please state "The authors have declared that no competing interests exist". 

3. Please amend your detailed Financial Disclosure statement. This is published with the article, therefore should be completed in full sentences and contain the exact wording you wish to be published.

ii). State the initials, alongside each funding source, of each author to receive each grant.

iii). State what role the funders took in the study. If the funders had no role in your study, please state: “The funders had no role in study design, data collection and analysis, decision to publish, or preparation of the manuscript.”

4. In the online submission form, you indicated that "The data that support the findings of this study are available from the corresponding author upon reasonable request following the icddr,b’s data sharing policy.". All PLOS journals now require all data underlying the findings described in their manuscript to be freely available to other researchers, either 1. In a public repository, 2. Within the manuscript itself, or 3. Uploaded as supplementary information.

Additional Editor Comments:

The study has important contributions to make because the global Covid-19 pandemic is a contemporary health condition is a public health concern of wide implication in all communities. The review of the manuscript has revealed important flaws in the conceptualization, theoretical and implementation of the problem phenomenon which must be corrected to position the thesis in the study to make the expected contribution to knowledge.

1. The title requires modifications to reflect what the study aimed to accomplish. The variables appearing in the body of the manuscript are not appropriately reflected in the title especially the explanatory variables. The impression is that knowledge and attitude are the only variables to explain the dynamics of Covid-19 transmission prevention in the population, whereas demographic characteristics are also involved. It is very important to determine how these variables were measured in terms of the validity of measures. Title may require some minor modification for clarity to “Knowledge, attitude, and practice of Bangladeshi urban slum-dwellers towards COVID-19 transmission-Prevention: A cross-sectional study”

2. The Abstract require some statement modification to establish clarity:

Background: The first COVID-19 case in Bangladesh was detected on 8th March 2020. Since then, efforts are ongoing across the country to raise awareness among the population to prevent the spread of this virus. We aimed to examine the knowledge, attitude, and practice of urban slum dwellers towards COVID-19 and determine to determine which of these factors most significantly predict COVID-19 Transmission-Prevention practice among the population.

The introduction of "factors" is confusing. What are these factors? Challenge to clarity of the aim of the study is the introduction of the word “factor” whereas, the knowledge, Attitude and antecedent factors required to understand the dynamics of the problem phenomenon in the transmission of Covid-19 within the population that require intervention of raising awareness. Which other factors are the authors referring to as “…Multiple regression models were applied to identify the associated factors with KAP scores” in line 30?

The abstract require revision.

3. Background to the study:

Line 53 -54 should read; thus, “…such as being in close proximity to an infected person or in an environment where droplets generated from coughs, sneezes or exhalation of an infected person, by touching a contaminated surface among others.”

It is pertinent in proposing what serves as the most effective alternative in the absence of a herd immunity to be conferred by vaccination of large proportion of the population as suggested in lines 54 – 64, to elucidate the theoretical basis for suggesting, in line 63, 64 that knowledge and attitude are significantly associated with the appropriate practice in the prevention and control of spreading COVID-19. This should be grounded in behaviour theories and public health principles of prevention. Unfortunately, this has not been done and very important variable to be considered, perception, has been missed out. The health-seeking paradigm of the of the health belief model (HBM) should have been applied here to validate the constructs used in the development of the instrument. It is noteworthy that public health research has evolved beyond the superficial approaches of the past lacking in grounded theories. The lack of these theories and epidemiological principles to elucidate the likely dynamics of the problem phenomenon to be diagnosed has weakened the study and invalidated any data derived.

The paragraph addressing this aspect is central to theoretical and conceptual clarifications that provides the desperately needed validity to the study, in the absence of which invalidates the entire study argument and thesis.

The authors are assuming that KAP represents a single construct as stated in the computation of sample size in line 96 “…50% of the adult slum population have proper KAP…”. This would be appropriate if the construct KAP is a weighted-aggregate score of the composite knowledge, Attitude and preventive practice regarding Covid-19 infection.

3. There is observed serious inconsistency in the statement of course of action to explore the likely predictors of the problem phenomenon. For instance, the study states that “An exploratory study design was used to assess the KAP of urban slum dwellers”, but no where did it state to “…identify the associated factors with KAP scores”. If the study is to identify factors associate with Covid-19 infection prevention, the factors mentioned would include all explanatory variables such as the demographic characteristics, moderating variables derived from the Health Belief Model guiding the design and implementation of the study. The computed 384 household should individuals per household.

Line 98 should read; “…estimated 384 consenting individuals/households were required to be enrolled for the study”.

Line 99 should read; “…512 individuals/households were selected by multi-stage sampling technique and data collected by telephone interviewer-administered methods with only 476 successful interviews.” Delete line 101.

Data collection instruments and measures:

It is not enough to mention that “…Instrument was developed by the research team based on published articles related to the assessment of KAP towards COVID-19 and the general guidelines recommended by the Bangladesh government and WHO”, Line 111-112, involves the development and validity of the instrument for data collection and needs to mention, not only published articles related to the assessment of KAP towards COVID-19, but the theoretical foundations of the constructs applied in an attempt to collect data to validate the thesis of factors associated with Covid-19 prevention practices of urban slum dwellers in Bangladesh.

For measures and item development, it is customary to have statement of attitude as items instead of questions and likewise with practice. Responses should be in the 4 response Likert-type scales with few examples illustrating actual statements in lines 124 – 130.

The structuring of the variables and measures are weak and flawed as can be seen in the statement of results. It would have been most appropriate to develop weighted aggregate scores and use these to generate results.

Attitudes are usually first measured as attitudinal dispositions and subsequently transformed as standardized scores for each participant in the study then frequency distribution into negative and positive attitudes. Practice is usually measured as frequency of practice/behaviour and has response categories of “not at all”, “really”, “occasionally” and always” and not “yes” or “no”.

4. As a result of the flawed measurements (Instrument for data collection, data collected, analysis) the results presented are not able to answer the research questions of what is the level of knowledge of participants in the study, their attitudinal dispositions, their attitudes and level of Covid-19 infection prevention practice?

5.By extension, discussion is observed as a re-statement of the results.

The observed weaknesses in the manuscript pointed out, if addressed adequately, would reposition the manuscript for possible publication.

Reviewers' comments:

Reviewer's Responses to Questions

**Comments to the Author**

1. Does this manuscript meet PLOS Global Public Health’s publication criteria? Is the manuscript technically sound, and do the data support the conclusions? The manuscript must describe methodologically and ethically rigorous research with conclusions that are appropriately drawn based on the data presented.

Reviewer #1: Yes

Reviewer #2: Yes

2. Has the statistical analysis been performed appropriately and rigorously?

Reviewer #1: Yes

Reviewer #2: Yes

3. Have the authors made all data underlying the findings in their manuscript fully available (please refer to the Data Availability Statement at the start of the manuscript PDF file)?

Reviewer #1: Yes

Reviewer #2: No

4. Is the manuscript presented in an intelligible fashion and written in standard English?

Reviewer #1: Yes

Reviewer #2: Yes

5. Review Comments to the Author

Reviewer #1: A) I think it's unfair to say poor knowledge at 48% in conclusion. You are referring to 21% poor as well.

B) Were participants incentivized in any form? Please mention.

C) 83% were 18-45 and 14% were between 46-60 years. What about remaining 3%. Are they above 60? Please clarify.

D) You have mentioned programatic recommendations. Do you have any research oriented recommendations? Would you recommend any other research areas within this context to get better understand the problem?

E) I was not able to see the tables in the submission

Reviewer #2: Manuscript is provided in a logical and concise fashion.

The only concern is the provision of data in line with the journal's policy. All data should be made fully available during submission of manuscript.

6. PLOS authors have the option to publish the peer review history of their article (what does this mean?). If published, this will include your full peer review and any attached files.

**Do you want your identity to be public for this peer review?** For information about this choice, including consent withdrawal, please see our Privacy Policy.

Reviewer #1: No

Reviewer #2: No

---

## [Editor Report · Decision Letter 1]

12 Jul 2022

PGPH-D-22-00360R1

Knowledge, attitude, and practice of Bangladeshi urban slum dwellers towards COVID-19 transmission-prevention: A cross-sectional study

Dear Md. Hasan,

Thank you for submitting your manuscript to PLOS Global Public Health. After careful consideration, we feel that it has merit but does not fully meet PLOS Global Public Health’s publication criteria as it currently stands. Therefore, we invite you to submit a revised version of the manuscript that addresses the points raised during the review process.

EDITOR' OBSERVATION: 

The authors' have revised most of the issues raised by the reviewers adequately but there are a few issues to consider.The manuscript has some editorial and technical correction required to strengthen the thesis presented and as much as possible address these observed weaknesses as listed.It is observed that the theoretical issues raised were not adequately addressed. Kindly review the validity of including this aspect in the manuscript.

We look forward to receiving your revised manuscript.

Kind regards,

Nnodimele Onuigbo Atulomah, PhD

Academic Editor

Journal Requirements:

Additional Editor Comments (if provided):

The manuscript presented has improved tremendously, having effected most of the recommended revisions. However, to strengthen the manuscript further, there are observed weaknesses that needs to be address. Most of these are editorial inadequacies but the theoretical issues defining the foundation of the thesis of the study still has flaws.

1. Title modification has been implemented.

2. In the abstract: Line 30-31 appear confusing. This should read thus; "A total of 80% or more score for each of KAP variables was classified as ‘good’, scores of less than 80% to 50%as moderate and lower than that as ‘poor’." for clarity.

Please revise line 162-163 to align appropriately.

Usually, the most appropriate method of categorization that is scientifically-based is the use of quartiles where the first quartile represents poor, the second and third represents below and above average while the fourth quartile represents good level. If you wish to combine the second and the third quartiles to represent average/moderate level, it is equally acceptable. How was attitude determined as good? Usually attitudes are classified as positive or negative, but initially a measure of the attitudinal disposition as a score of the sample in the study is derived and standardized using the Z-score which identifies individuals in the sample distribution with negative attitudes and those with positive attitudes. Therefore, the computations resulting to information expressed in line 35 should be correctly computed.

Background to the Study:

Most are well expressed but three editorial issues need resolving;

1. Line 84-85 would best be rendered as "Studies showed that knowledge regarding COVID-19 and prevention measures to be taken, and attitudinal dispositions are significantly associated with appropriate infection prevention practice, which has the potential of playing significant role in the prevention and control of community transmission of COVID-19 [20]"

2. Line 98 would better be rendered as, "Evidence from studies showed that…" as three studies have been cited-(30-32)

3. Again lines 102-104 would benefit from expressing the significance of the study as; "Therefore, it is important to assess the KAP of slum population towards COVID-19, as a rapid assessment, to evaluate levels of existing knowledge gap and provide basis for policy brief to inform policymakers in the design and implementation of public interventions required in slum areas."

Materials and Methods require minor editorial corrections:

1. Study Design needs to be appropriately expressed. Line 109-111 should be recast to read, thus; "This was a cross-sectional telephone-based exploratory study conducted among 476 adult individuals (>18 years) of 476 households from the selected slums during 31 October to 1 December 2020."

Note line 115 and 144, the acronym for "International Centre for Diarrhoeal Disease Research Bangladesh" should be in upper case letters.

2. The questionnaire development is the major area of concern in the manuscript:

Line 144 -146 failed to indicate the theoretical foundations of the conceptual framework for the study as pointed out in the editor's comments for revision, "Data collection instruments and measures:

It is not enough to mention that “…Instrument was developed by the research team based on published articles related to the assessment of KAP towards COVID-19 and the general guidelines recommended by the Bangladesh government and WHO”, Line 111-112, involves the development and validity of the instrument for data collection and needs to mention, not only published articles related to the assessment of KAP towards COVID-19, but the theoretical foundations of the constructs applied in an attempt to collect data to validate the thesis of factors associated with Covid-19 prevention practices of urban slum dwellers in Bangladesh.

Importantly, the implications of lack of adequate and properly developed theoretical and conceptual clarifications of the constructs in the study has resulted in errors of interpretations leading to inappropriate items developed for especially attitudinal dispositions of the participants in the study. All the 10-item did not represent attitudinal dispositions. In these item, there was observed inability to distinguish what constitutes an attitude and perception, to facilitate appropriate operationalization of attitude in the study. "Attitudes refers to emotional responses expressed emerging from mental dispositions triggered by external circumstances in the environment." Furthermore, the response pattern for attitude and preventive-practice in the instrument are seriously faulty for the constructs. The likert-type response pattern is usually applied depending of the nature of the phenomenon being considered. For example, practice takes into consideration four response-types of "not at all", "occasionally", "often" and "always"

Take for instance the items listed for practice of COVID-19 Prevention should not be questions but statements as it appears in the instrument but -("I sneeze into my elbow" with the response of [Not at all]; [Occasionally]; [Often] and [always]" these assess the relative degrees of compliance to preventive requirement and would be scored on the following weighted score of 0, 1, 2, 3"). The second need not have "often" in the statement.

Do you sneeze into your elbow?

Do you often touch your mouth, eye, and nose?

Do you maintain social distance when you go outside?

Do you wear mask when you go outside?

Data Analysis:

Kindly review line 167 for the acronym of Analysis of Variance and elsewhere this appears such as the tables. This should be One-way ANOVA.

Line 166-169 would better rendered as, "Both descriptive and inferential statistical analysis were performed with the Two tailed t-test for means of variables of two independent samples and the One-way ANOVA for variables of more than two independent samples to test the hypothesis of a statistical difference for each dimension of KAP scores with demographic and socioeconomic categories, separately."

Results has minor revision of editorial nature:

It is never considered appropriate to start the presentation of result with "Table 1 shows…", rather best practice is to begin with the actual result as a reported narrative of important results; very good and poor results, the reader is referred to the tables for other results.

Suggested approach for result presentation for lines 194-197; "The results in the study showed that a total of 476 respondents from 476 households who participated were heads (65%) of households with 82% being adults (18-45 years of age), 18% were older.

End the paragraph in line 202 with "(See Table 1)"

Revise line 207 accordingly, and line 255, 268, 296 among others like these.

Discussion:

Discussion narrative is best started with a brief mention of the objectives of the study to facilitate linking the thesis of the study with the findings and provide coherence of thoughts.

Kindly begin line 322 thus; "The study sought to assess the KAP of selected slum dwellers in Dhaka city, Bangladesh towards COVID-19 and found that a small proportion of the sample surveyed demonstrated good knowledge regarding the infection..."
---

## [Editor Report · Decision Letter 2]

15 Aug 2022

Knowledge, attitude, and practice of Bangladeshi urban slum dwellers towards COVID-19 transmission-prevention: A cross-sectional study

PGPH-D-22-00360R2

Dear Dr. Hasan,

We are pleased to inform you that your manuscript 'Knowledge, attitude, and practice of Bangladeshi urban slum dwellers towards COVID-19 transmission-prevention: A cross-sectional study' has been provisionally accepted for publication in PLOS Global Public Health.

Best regards,

Nnodimele Onuigbo Atulomah, PhD

Academic Editor

Congratulations, after reviewing the manuscript PGPH-D-22-00360R2, it has demonstrated that all relevant revisions recommended have been made and therefore, I am recommending to the Editor-in-Chief to accept the manuscript for further editorial process towards publication.